# Epitaxial growth of an atom-thin layer on a LiNi$_{0.5}$Mn$_{1.5}$O$_4$ cathode for stable Li-ion battery cycling

Xiaobo Zhu [1,2], Tobias U. Schülli[1,3 ✉], Xiaowei Yang[4], Tongen Lin [1], Yuxiang Hu[1], Ningyan Cheng[5], Hiroki Fujii[6], Kiyoshi Ozawa[6], Bruce Cowie[7], Qinfen Gu [7], Si Zhou [4,5], Zhenxiang Cheng [5], Yi Du [5] & Lianzhou Wang [1 ✉]

Transition metal dissolution in cathode active material for Li-based batteries is a critical aspect that limits the cycle life of these devices. Although several approaches have been proposed to tackle this issue, this detrimental process is not yet overcome. Here, benefitting from the knowledge developed in the semiconductor research field, we apply an epitaxial method to construct an atomic wetting layer of LaTMO$_3$ (TM = Ni, Mn) on a LiNi$_{0.5}$Mn$_{1.5}$O$_4$ cathode material. Experimental measurements and theoretical analyses confirm a Stranski–Krastanov growth, where the strained wetting layer forms under thermodynamic equilibrium, and it is self-limited to monoatomic thickness due to the competition between the surface energy and the elastic energy. Being atomically thin and crystallographically connected to the spinel host lattices, the LaTMO$_3$ wetting layer offers long-term suppression of the transition metal dissolution from the cathode without impacting its dynamics. As a result, the epitaxially-engineered cathode material enables improved cycling stability (a capacity retention of about 77% after 1000 cycles at 290 mA g$^{-1}$) when tested in combination with a graphitic carbon anode and a LiPF$_6$-based non-aqueous electrolyte solution.

[1] Nanomaterials Centre, School of Chemical Engineering, and Australian Institute of Bioengineering and Nanotechnology, The University of Queensland, Brisbane, QLD 4072, Australia. [2] College of Materials Science and Engineering, Changsha University of Science and Technology, Changsha 410114, China. [3] ESRF—The European Synchrotron, 38000 Grenoble, France. [4] Key Laboratory of Materials Modification by Laser, Ion and Electron Beams (Dalian University of Technology), Ministry of Education, Dalian 116024, China. [5] Australian Institute for Innovative Materials (AIIM), University of Wollongong, Squires Way, North Wollongong, NSW 2500, Australia. [6] National Institute for Materials Science, 1-2-1 Sengen, Tsukuba-city, Ibaraki 305-0047, Japan. [7] Australian Synchrotron, 800 Blackburn Road, Clayton, VIC 3168, Australia. ✉email: schulli@esrf.fr; l.wang@uq.edu.au

Epitaxy refers to the guided growth of a crystal on a crystalline substrate, where the atoms in an epitaxially growing layer are in registry with the atoms in the underlying substrate. Owing to the well-defined structure/interface or selectively exposed facets, epitaxially grown thin films and nanostructures have exhibited outstanding performance in various applications, including electronics, optoelectronics, and catalysis[1,2]. As the properties of surfaces and interfaces are crucial in many other materials systems and applications, such as metal oxides for rechargeable batteries[3], epitaxially engineered materials can possibly solve critical problems in these areas.

Lithium ion batteries (LIBs) are of vital importance in consumer electronic, electric vehicle, and energy storage sectors. The increasing pressure for decarbonization pushes the development of LIBs towards lower cost, higher energy density, and longer cycle life. The performance of LIBs largely depends on the evolution of surfaces and interfaces of the battery materials[3,4]. This is particularly critical for high-voltage cobalt-free cathodes, represented by the 5 V-class spinel $LiNi_{0.5}Mn_{1.5}O_4$ (LNMO)[5–8], which is promising for its low cost and high energy density but suffers rapid degradation in full batteries due to the severe interfacial reactions between its Jahn–Teller (J–T) distorted surface lattices and the electrolyte species[9–12]. In this regard, surface passivation strategies, such as wet-chemistry coatings[13–15], atomic layer deposition[16–18], and the use of binders with tailored properties[19] and film-forming electrolyte additives[20–23], have been extensively explored to reduce the interfacial reactions. However, most passive layers were formed under thermodynamic non-equilibrium at low temperatures (typically below 500 °C) to avoid interdiffusion[13–18], resulting in surface deposits that are not bonding to the host lattices. Meanwhile, the low temperatures could not guarantee surface diffusion, the passive layers were conventionally prepared to have sufficient thickness (from several to dozens of nanometres) to guarantee better coverage. Being considerably thick and crystallographically disconnected with the host lattices, these layers could detach from the dynamic Li hosts easily and introduce significant charge transfer barriers as well[24]. Consequently, these issues limit the improvement of the cycle life of high-voltage cobalt-free cathode materials in full Li-ion cells.

In this work, we apply the knowledge of epitaxy to the development of a stable passivation layer of monoatomic thickness on the LNMO cathode materials by a simple mixing-and-calcination method. The immiscibility of La in the LNMO substrate together with the energetic advantage of La–O surface terminations drive the formation of a unique surface layer under thermodynamic equilibrium, which has been tracked down by theoretical calculations, quantitative X-ray analyses, and other methods independently. Furthermore, the epitaxially grown cubic $LaTMO_3$ (TM = Ni, Mn) has a lattice mismatch of ca. 5 % with the LNMO substrates, leading to Stranski–Krastanov type of growth, where a strained wetting layer is formed and self-limited to monolayer thickness prior to the nucleation of the relaxed islands (Fig. 1a). When tested in full Li-ion coin cell configuration, the coated LNMO in combination with a graphite anode and a $LiPF_6$-based non-aqueous electrolyte solution, enable a capacity retention of about 77% after 1000 cycles at 290 mA g$^{-1}$ and a final discharge capacity of about 80 mA h g$^{-1}$ with an average coulombic efficiency (CE) > 99%.

## Results

### Growth of $LaTMO_3$ (TM = Ni, Mn) on $LiNi_{0.5}Mn_{1.5}O_4$.
Density functional theory (DFT) calculations have been carried out to theoretically understand the structure and stability of atomic layer La–TM–O on the surface of LNMO. A single atomic layer of La–Ni–O was placed on the (111) surface of LNMO. Simulation optimization reveals that such atomic layer consisting of La: Ni: O in a ratio of 1: 2: 3, where the surface unit cell reaches into the first TM layer and is hence referred to as $LaNi_2O_3$, indicating a high degree of integration between the surface structure and the underlying lattice. As displayed in Fig. 1b, each Ni atom in the two-dimensional layer is coordinated with three O atoms. The underneath La atoms are bonded with the O atoms from the substrate surface, with interlayer spacing of 0.359 nm (0.190 nm) from the topmost Ni (La) atoms to the substrate surface. Differential charge density (Fig. 1c) shows charge accumulation in the interfacial region with electron transfer from La atoms to the surface O atoms of the substrate. In addition, according to the electronic density of states in Fig. 1d, the monolayer $LaNi_2O_3$ on the surface of LNMO exhibits metallic behavior with prominent states at the Fermi level, beneficial for electron conduction. The change of the surface TM does not alter the stability of the heterostructure (Supplementary Fig. 1).

To verify the concept of self-limiting growth of the passive layer, we have carried out a systematic study on the intake of different dopants into the LNMO host lattice and analysed the impact on the structure, the solubility limit of dopants, and the eventual existence of secondary phases. The particular effects of La integration in the bulk lattice of LNMO are presented in Fig. 1, representing two routes: a "doping" process, which is sintering the pre-mixed sol-gel precursors at 750 °C, produces LNMO-D0.3La, LNMO-D0.5La, LNMO-D0.7La, and LNMO-D1.0La, corresponding to products containing 0.3 to 1.0 at% La with respect to total TM (Fig. 2a–c); and a "coating" procedure, which is re-heating the mixtures of crystalline LNMO and La salt of same atomic ratios, generates LNMO-C0.3La, LNMO-C0.5La, LNMO-C0.7La and LNMO-C1.0La (Fig. 2d–f). The detailed synthesis methods are shown in the Methods section. Synchrotron X-ray powder diffraction (XRD) patterns of all the products correspond to the expected spinel structure with a cubic lattice parameter of $a = 0.817$ nm. In the case of total or partial miscibility, interdiffusion leads to similar structural results for most probed dopants (Al, Ga, and Y) for both processes (Supplementary Figs. 2 and 3). The analysis of crystal quality and size in Supplementary Fig. 4 underlines that La as a dopant presents an exception: beyond poor miscibility in LNMO, La doping reduces the crystal quality dramatically over sintering, indicating that the diffusion of the large La ions in the lattice encounters important barriers. The crystal degradation of LNMO led by La doping is also visible from the scanning electron microscopy (SEM) images (Supplementary Fig. 5). The immiscibility of La in LNMO can be explained by the larger ionic size of La$^{3+}$ (0.1032 nm compared to 0.053 nm for Mn$^{4+}$ and 0.069 nm for Ni$^{2+}$) as well as the much stronger La–O bond (the bond dissociation energies of La–O, Mn–O, Ni–O are around 798, 362 and 366 kJ mol$^{-1}$)[25]. This is also consistent with the surface presence of La$^{3+}$ in Ni-rich layered oxides[26]. In our case, the immiscibility of La permits sufficient surface diffusion for better coverage at high temperatures without the concern of interdiffusion.

As can be seen in Fig. 2b, e, secondary $LaTMO_3$ perovskite phases emerge when La concentrations reach and surpass 0.7 at% and 0.5 at% for doping and coating processes, respectively. The observed secondary XRD peaks can be identified as a rhombohedral ($R\bar{3}c$) phase (Fig. 2c) in the doping route, which is a common symmetry for $LaTMO_3$ due to J–T distortion[27]. Interestingly, a cubic ($Pm\bar{3}m$) (Fig. 2e) symmetry is found in the coating products. As atoms on free surfaces often tend to form surface specific structure in the case of immiscibility[28], here the cubic LNMO substrate directs the formation of an epitaxial $LaTMO_3$

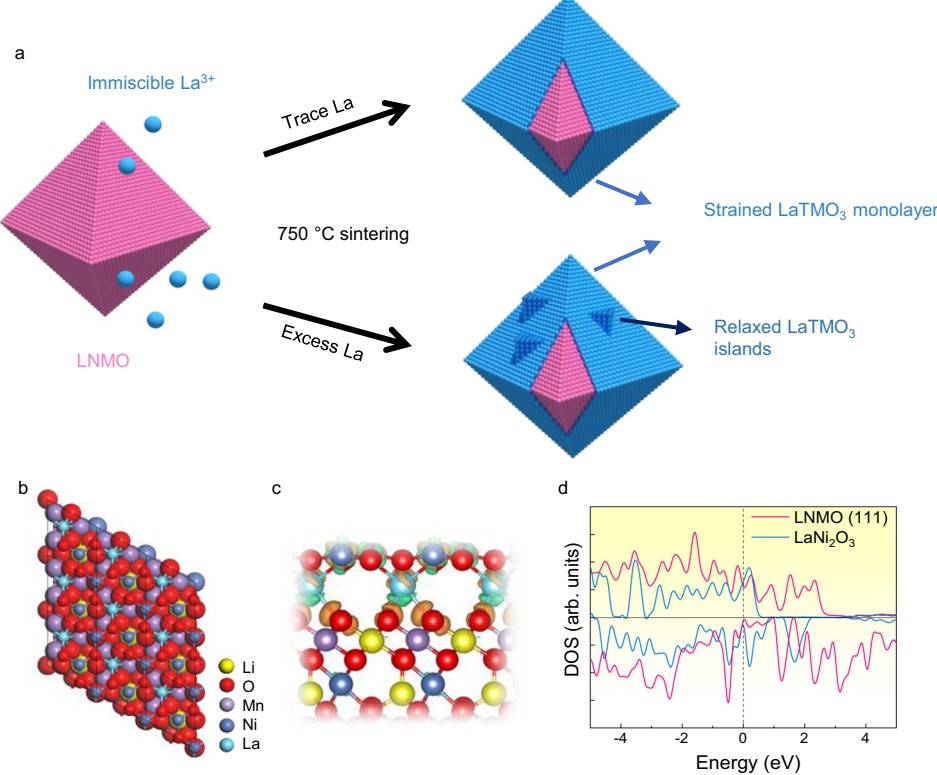

**Fig. 1 The integration behaviour of La with the lattice of LNMO. a** Schematic of the growth of LaTMO$_3$ on the facets of LNMO. **b** Model structures (top view), **c** differential charge density (side view), and **d** spin-polarized density of states (DOS) of monolayer LaNi$_2$O$_3$ on LNMO (111) substrate by DFT calculations. The black boxes in **b** indicate the lateral dimension of the supercell. The orange and green colours in C represent electron accumulation and depletion regions, respectively, with isosurface value of 0.007 $e$/Å$^3$. The dashed line in **d** shows the Fermi level that is shifted to zero.

film with biaxial strain[29]. In fact, the low index facets of both secondary phases of LaTMO$_3$ present a strong crystallographic similarity to the {111} facets of the octahedral spinel crystals (Supplementary Fig. 6). Furthermore, the cubic LaTMO$_3$ even possesses epitaxial match with the truncated {100} facets of LNMO octahedra (Supplementary Fig. 7). The J–T distortion-free LaTMO$_3$ epitaxial film could be a desirable passive layer to reduce the vulnerability of the LNMO surface lattice towards the electrolyte.

**Quantification of the LaTMO$_3$ (TM = Ni, Mn) secondary phase on LiNi$_{0.5}$Mn$_{1.5}$O$_4$.** Quantitative X-ray methods are well established as a basic tool in crystallography. Here we use the quantitative X-ray analysis to determine the amount of material contributing to the scattering process[30–32]. The normalization on the LNMO peaks allows for a direct quantification of the La fraction presented in the LaTMO$_3$ secondary phases, and is also useful to extract independently the nucleation threshold because the difference between the precursor fraction of La and the fraction in the LaTMO$_3$ can be determined from the relative integrated XRD intensities. The details to derive the quantities from the relative intensities of the diffraction peaks are disclosed in the Supplementary Note 1 (see Supplementary Information). As shown in Fig. 3a, the increase of the integrated LaTMO$_3$ intensity as a function of La content in the precursor mixture can be fitted by linear regression to obtain an estimate for the fraction of La that remains absent from the secondary phase. The intersection with the *X*-axis yields the amount of La missing from the secondary phase or the threshold value of La adsorbed at the surface prior to nucleation. For monolayer coverage of a regular octahedron with side length *l*, the ratio between surface atoms $N_S$

and bulk atoms $N_V$ can be expressed as $\frac{N_S}{N_V} = \frac{S*\rho_S}{V*\rho_V} = \frac{3\sqrt{6}}{l}\frac{\rho_S}{\rho_V}$, with $\rho_S$ and $\rho_V$ being the respective surface and volumetric densities of atoms counting only TM and La. For a LNMO lattice parameter of 0.817 nm, this results in $\rho_V = 0.000029$ nm$^{-3}$ and $\rho_S = 0.00034$ nm$^{-2}$ for the (111) facet and we obtain $\frac{N_S}{N_V} = \frac{0.88 \text{ nm}}{l \text{[nm]}}$. This hyperbolic relationship between the fraction of surface atoms and the average side length of an octahedral particle is plotted in Fig. 3b. Colored lines mark the positions represented by the fitted data from Fig. 3a. The monolayer coverage is also verified by investigating larger-sized LNMO crystals (BLNMO, average size of 900 nm), which are prepared by sintering the precursor at a higher temperature of 900 °C. The determined amount of La adsorbed by the surface is well in line with the observed mean crystal size as seen in the SEM images (insets of Fig. 3b).

The surface presence of La in these samples is also supported by spectroscopic characterizations. Figure 3c shows near edge X-ray absorption fine structure (NEXAFS) spectra for a doping precursor containing 0.5 at% La, LNMO-D0.5La, and LNMO-C0.5La. Considering an approximate exponentially damped probing depth of 20 nm in total electron yield (TEY) mode[33] and a particle radius of 80 nm, the significant increase of the La signal in the coated sample confirms the surface presence of La. Furthermore, depth-profiling X-ray photoelectron spectroscopy (XPS) was applied to the LNMO-C0.5La. Under Ar$^+$ etching, the gradual disappearance of the characteristic La 3$d_{5/2}$ double peaks from the XPS spectrum of the LNMO-C0.5La are shown in Fig. 3d. The La peaks disappear at the onset of etching while no impact on the Ni-peaks is observed, confirming the surface enrichment of La.

Interestingly, the LaTMO$_3$ islands predicted are conspicuous in the SEM image of LNMO crystals coated by excess La of 2 at%

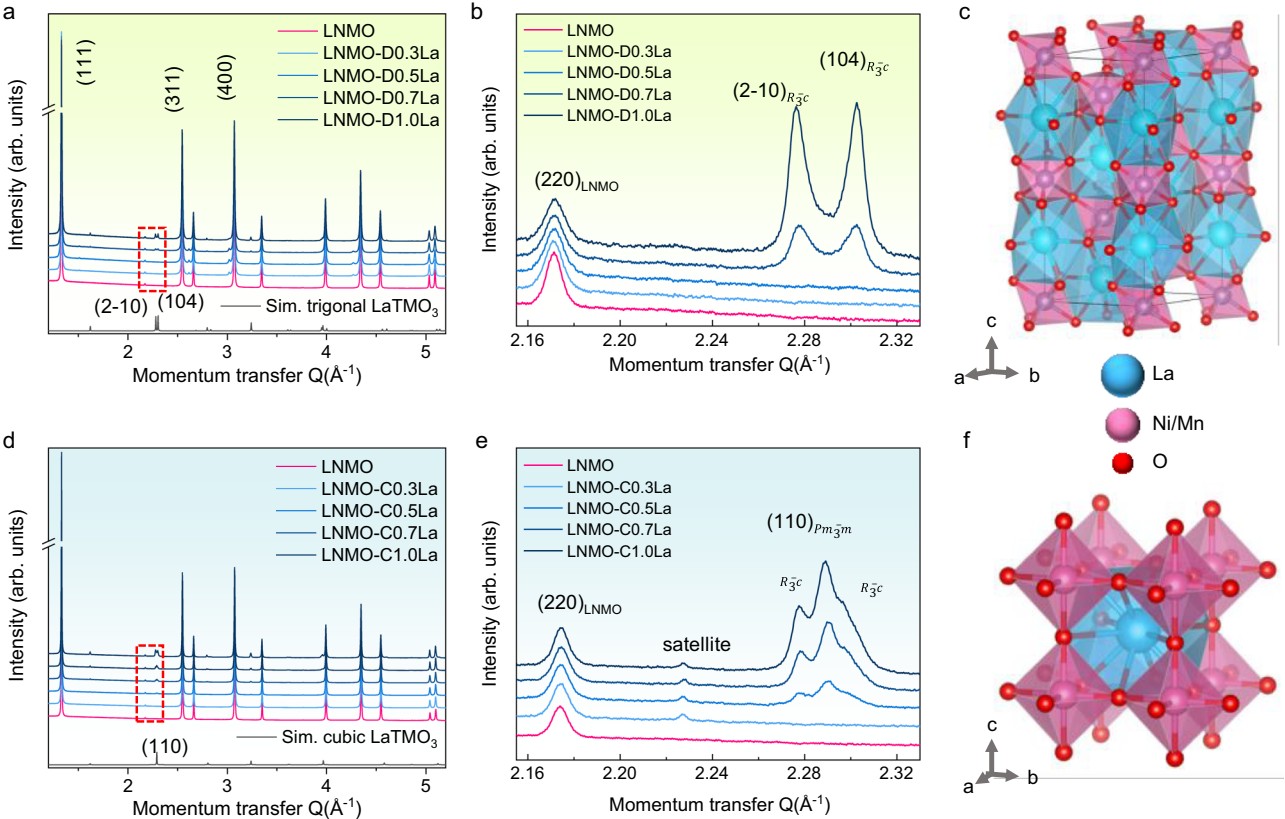

**Fig. 2 Phase analysis of the LNMO samples modified by La. a** Synchrotron XRD data of pristine and La doped LNMOs. **b** Zoom of the dashed red frame in (**a**) showing the characteristic LaTMO$_3$ peaks in doped samples. **c** The structure model of rhombohedral LaTMO$_3$. **d** Synchrotron XRD data of pristine and La coated LNMOs. **e** Zoom of the dashed red frame in **d** indicating the characteristic LaTMO$_3$ peaks in coated samples. **f** The structure model of cubic LaTMO$_3$.

(Supplementary Fig. 8). For the BLNMO crystals, dense islands are observed at only 1 at% La (BLNMO-C1.0La, Supplementary Fig. 9). High-angle annular dark-field scanning transmission electron microscopy (HAADF-STEM) image (Fig. 3e) and corresponding energy dispersive spectroscopy (EDS) elemental mappings (Fig. 3f) not only indicate a good coverage of La on the LNMO crystal, but also confirm that these surface islands are La-rich domains. The compositional difference between the island and the bulk has been further verified by the EDS quantitative analysis (Supplementary Fig. 10 and Supplementary Table 1). Figure 4a shows a high-resolution transmission electron microscopy (HR-TEM) image of the LNMO-C0.5La taken from [110] axis. The inner lattice planes with inter-spacings of 0.47, 0.47, and 0.41 nm correspond to (-111)$^{1-11}$, and (002) of the spinel phase (Fig. 4b). Different from the smooth edge in the pristine LNMO (Supplementary Fig. 11), LNMO-C0.5La shows a discontinued outer layer, which can be attributed to 1–2 layers of LaTMO$_3$ as the predicted structural model (Fig. 4c). HAADF-STEM image (Fig. 4d, e) captured at the boundary between LNMO and LaTMO$_3$ islands unveil the atomic configuration of the heterostructure. The cubic LaTMO$_3$ epitaxially grows on cubic LNMO by sharing the (111) plane with clear dislocations, consistent with the structure prediction (Fig. 4f). The annular bright-field scanning transmission electron microscopy (ABF-STEM) images taken at the edge of LNMO-C0.5La and at the boundary between the spinel and the La-rich surface islands are shown in Supplementary Fig. 12. In combination with the XRD analysis, the crystal growth mode of wetting and islanding can be further verified.

Considering the above-mentioned experimental and computational findings, this epitaxial monolayer coverage is expected to

favorably form in such a system: being compositionally immiscibly in yet structurally compatible with the LNMO substrate, the cubic LaTMO$_3$ offers a La–O bond rich surface termination for the LNMO. The gain in energy of this construction (0.22–0.29 J/m$^2$, confirmed by our DFT model) is superior to the elastic energy of 0.16 J/m$^2$, which is required to strain by 5% one monolayer of LaTMO$_3$. For epitaxial growth of LaTMO$_3$ beyond one monolayer, the energetic advantage of La–O surface terminations is no longer relevant and the high lattice mismatch, as well as the low LaTMO3 (100) surface energy, leads to the onset of islands[34]. As calculated in the Supplementary Information and presented in Supplementary Fig. 13, islands as small as 8 nm are thermodynamically stable. This island formation process is a textbook example for the onset of Stranski–Krastanow growth that ultimately leads to relaxed island nucleation on top of a strained wetting layer of monoatomic thickness[35]. The satellite peak presented halfway in between the LNMO (220) and the cubic LaTMO$_3$ (110) peak in Fig. 2e is also a clear evidence of the epitaxial relationship with mismatch relaxation indicating a dislocation network for coverage higher than one atomic layer [36].

**Li-ion storage properties of the epitaxially-engineered LiNi$_{0.5}$Mn$_{1.5}$O$_4$ cathode.** To verify the effectiveness of the surface passivation, the electrochemical performance of the untreated and treated LNMOs has been carefully compared under the same experimental conditions including the use of LiDFOB as an electrolyte additive[37], which is found necessary for the extended cycling of LNMOs (Supplementary Fig. 14). When applied in coin-type cells using Li foils as the counter electrodes, all LNMO cathodes exhibit dominant redox plateau at around 4.7 V (vs. Li/Li

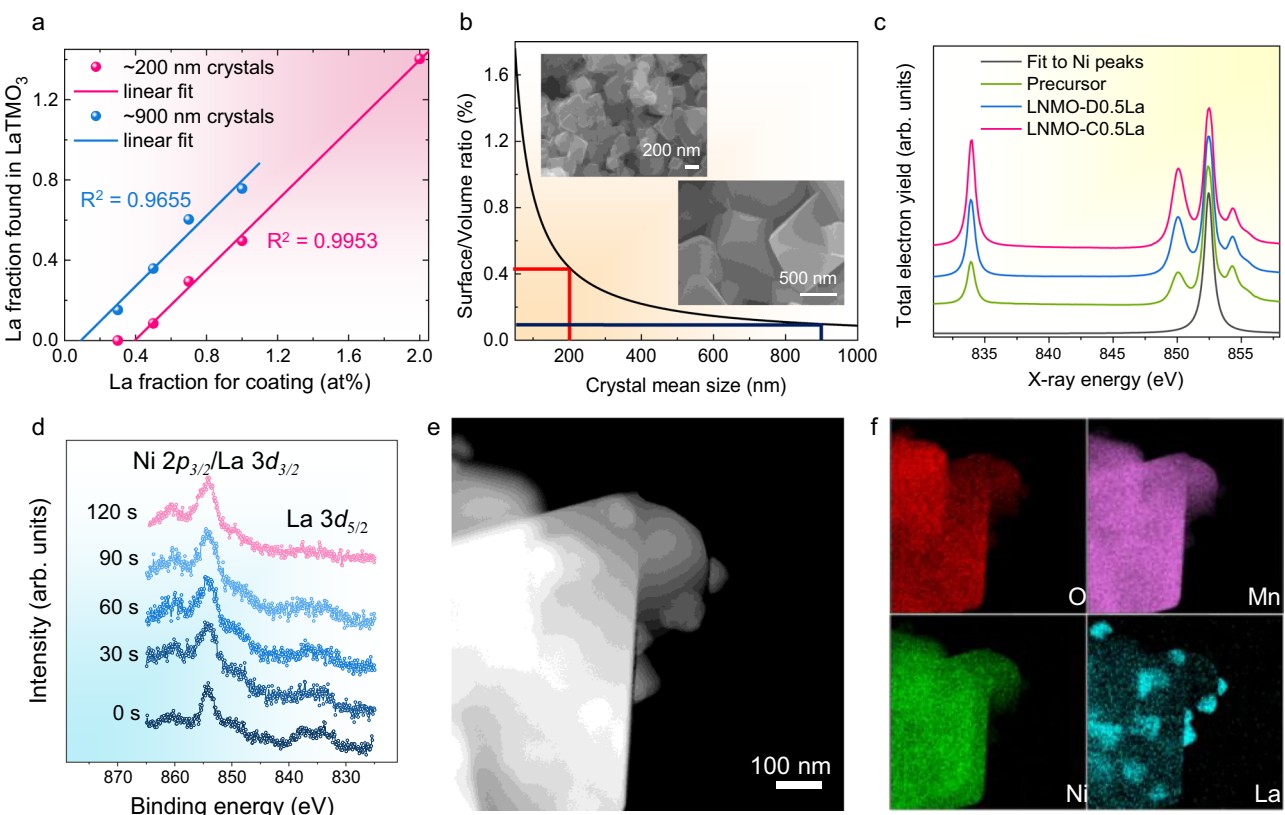

**Fig. 3 Surface enrichment of La on the LNMO samples. a** Integrated normalized intensities of LaTMO$_3$ peaks as a function of relative La concentration, normalized on the LNMO (311) peak. The fitting errors ($R^2$) are 0.9953 and 0.9655 for ~200 and ~900 nm LNMO crystals. **b** Surface to volume ratio as a function of the size of the LNMO octahedra. The surface adsorption fraction depends on the size of the LNMO octahedra shows a shift to higher values for smaller crystals (inset shows the SEM images). Coloured lines indicate the points on this curve as extracted from the data in (**a**). **c** NEXAFS spectra of La and Ni signals showing increasing surface presence of La for La doping precursor, LNMO-D0.5La, and LNMO-C0.5La (the integrated intensity is normalized to the Ni peak). **d** XPS spectra of LNMO-C0.5La etched by Ar$^+$ etching as a function of etching time. **e** HAADF-STEM image of a typical BLNMO-C1.0La. **f** Corresponding elemental mappings of (**e**).

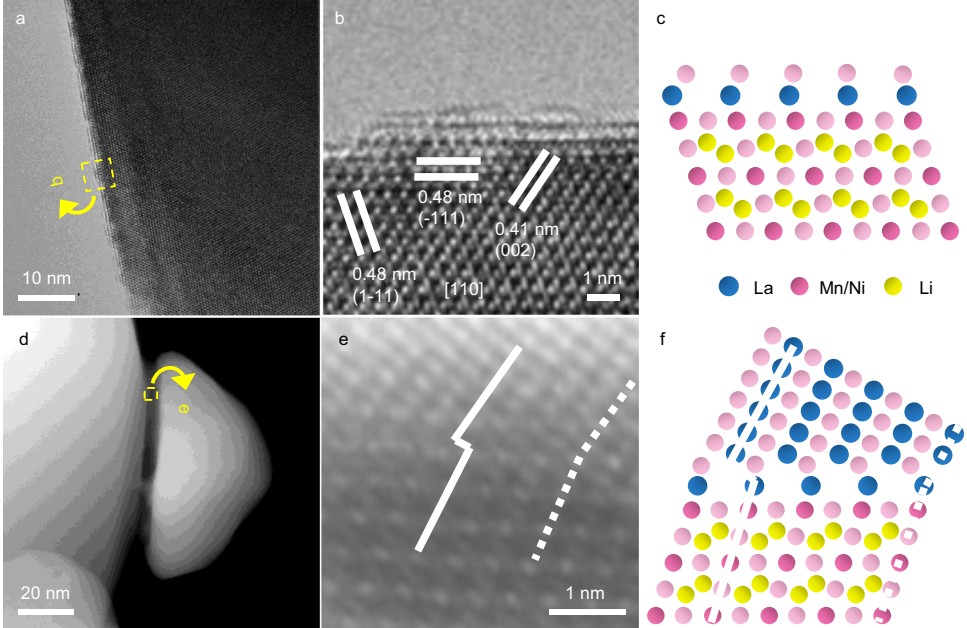

**Fig. 4 TEM analysis on the LNMO-LaTMO$_3$ heterostructure. a** HR-TEM image captured at the edge of LNMO-C0.5La from [110] axis. **b** Zoom on the marked region in (**a**) (rotated). **c** Predicted structural model of LNMO-C0.5La. **d** HAADF-STEM image taken at the boundary between the spinel and the La-rich surface island. **e** Zoom (rotated) on the interface showing interface defects (full vertical lines) and coincidences (dashed vertical lines). **f** The associated structural model of the LNMO-LaTMO$_3$ heterostructure.

[+]) (Supplementary Fig. 15a). The initial CEs are 77.8, 77.6, and 80.4% for LNMO, LNMO-D0.5La, and LNMO-C0.5La, respectively, indicating the LaTMO$_3$ coating can reduce the parasitic reactions between electrodes and electrolyte. All the LNMOs show good rate capability and long cycling stability in half-cell configuration with the LiDFOB electrolyte additive at 26 °C (Supplementary Fig. 15b, c). This behaviour is attributed to the three-dimensional spinel framework which offers fast ion diffusion and experiences a minimal structural fatigue over cycling[38–41]. Moreover, the atomically thin LaTMO$_3$ coating does not compromise the rate performance of LNMO due to the monolayer thickness and electronic conducting nature. However, an excessive amount of La (LNMO-C0.7La) affects the rate performance (Supplementary Fig. 16), which may be due to the thickening of relaxed LaTMO$_3$ islands that hinder the charge transfer.

As mentioned above, a major challenge preventing the practical application of LNMO is its poor cycling performance in full Li-ion cell configuration due to the TM dissolution. We then verified the effectiveness of this surface protecting strategy by pairing the LNMO cathodes with mesocarbon microbeads (MCMB) graphite anodes to assemble full cells. The charge/discharge curves of the MCMB∥LNMO and MCMB∥LNMO-C0.5La full cells are illustrated in Fig. 5a, b, which give an initial CE of around 60%. As shown in Fig. 5c, the MCMB∥LNMO-C0.5La full cell exhibits capacity retention of 77% over 1000 cycles for a high cell voltage cut-off of 4.8 V at 26 °C, much higher than that of 32% for the cell using LNMO cathode. Since the long cycling tests have been measured at a relatively high specific current and the addition of 0.2 M LiDFOB increases the concentration of electrolyte, the cell formation could take a long time in terms of the number of cycles[42]. This could explain the slight capacity increment of LNMO-C0.5La over cycling, where the TM dissolution is prevented. The performance of the full cells is also supported by the evolution of electrochemical impedance spectroscopy (EIS) spectra (Fig. 5d, e). According to the fitting results (Supplementary Fig. 17 and Supplementary Table 2)[43], the surface film resistance ($R_{sf}$) and charge transfer resistance ($R_{sf}$) of the standard MCMB∥LNMO cell is 1.4 and 1.7 times higher than those of MCMB∥LNMO-C0.5La cell after the first cycle. Remarkably, the differences are enlarged to 3.1 and 2.0 times after 1000 cycles. The substantial increase of resistance of the cell using pristine LNMO is aligned with the degradation mode led by TM dissolution, where the TM ions deposit on the graphite anode to stimulate the growth of a thick SEI layer[11]. Using the same synthetic method applied for La, we have also tested Al, Ga, and Y in LNMO. Unfortunately, we noticed that these elements are not stable on the surface of the cathode active materials and instead tend to dope into the LNMO lattice because of their miscibility with LNMO. As a result, LNMO-D0.5La and LNMOs treated with these alternative ions show worse performances in terms of full Li-ion cell cycling stability (Supplementary Fig. 18). As discussed in the previous sections, the BLNMO crystals require only ~0.1 at% La for a mono-atomic layer coverage. The BLNMO-C0.1La has also been prepared and evaluated in a full cell configuration, of which its capacity is even slightly increased after 1000 cycles at 26 °C (Supplementary Fig. 19). The extended capacity increase could be related to the bigger crystallite size as well as the effective surface protection[44]. The cycling stability of the MCMB∥BLNMO-C0.1La full cell has been further evaluated at 45 °C. The full cell retains 91 and 72% of its capacity after 500 cycles and 1000 cycles, respectively (Supplementary Fig. 20). Note that the weight percentage of La only accounts for 0.15 wt% of the cathode material for BLNMO-C0.1La, indicating the improved performance at minimal weight increase. It should be pointed out that the improved stability of the full cell is realized using a conventional LiPF$_6$-containing carbonate-based electrolyte solution, thus avoiding strategies like the use of highly concentrated or ionic liquid-based non-aqueous electrolyte solutions[45,46] which can be considered still not fully scalable for a wide adoption in practical electrochemical energy storage applications.

## Discussion

The long-term cycling performances disclosed in the previous section confirm the effective stability role of the LaTMO$_3$ wetting layer. As indicated by the ex situ XRD pattern of LNMO-C0.5La (Fig. 6a), the epitaxially grown LaTMO$_3$ is robust to survive after 1000 cycles. Ex situ post-mortem Raman measurements (Fig. 6b) on the negative electrodes after 1000 cycles reveal reduced $I_D/I_G$ for the graphite anode paired with LNMO-C0.5La cathode, showing less structural disordering when the opposite electrode is protected by LaTMO$_3$. Correspondingly, ex situ post-mortem EDS spectra (Fig. 6c, d) confirm much intensified signals of Mn from the cycled graphite anode coupled with pristine LNMO after 1000 cycles. The TM deposition and its effect in changing the morphology of the graphite anode are also tracked by the secondary and backscattered electron images (Supplementary Fig. 21). The amount of dissolved TM has been further quantified by inductively coupled plasma mass spectrometry (ICP-MS) measurements of negative electrodes harvested from cycled half-cells at charged state (Supplementary Fig. 22). The analysis confirms that the ultrathin LaTMO$_3$ wetting layer continuously reduces the TM deposition on Li electrodes. Ex situ post-mortem XPS characterizations of the electrodes after 100 electrochemical cycles (Supplementary Figs. 23–25) confirm the pronounced lattice oxygen signal in the cycled LNMO-C0.5La cathode as well as weaker Mn 2$p$ and Li 1$s$ signals from its paired graphite anode[47]. Relating these characterizations to the full cell performances reveals the role of the La modification. Under the conditions beneficial for thermodynamic equilibrium, La, immiscible in bulk LNMO, forms into a stable surface phase with epitaxial match to the host cathode crystal, guaranteeing efficient charge transfer as well as effective suppression of TM dissolution for stable 5 V-class battery (Fig. 6e).

In summary, epitaxy has been used to develop atomically thin and thermodynamically stable surface passivation on 5 V-class LNMO spinel crystals. The immiscibility of La in LNMO, energetical advantage of La–O terminations, together with the lattice mismatch led to the Stranski–Krastanov growth of LaTMO$_3$ wetting layer when La is introduced by the coating approach. In consistence with the growth mode, quantitative X-ray methods confirm that 0.5 at% La is required for the appropriate surface coverage of LaTMO$_3$ wetting layer on the LNMO octahedra. This explains the improved electrochemical performance of the LNMO-C0.5La compared with pristine LNMO, La-doped LNMO (LNMO-D0.5La), and LNMOs coated by insufficient (LNMO-C0.3La) or excessive LaTMO$_3$ (LNMO-C0.7La). The structural monolayer of LaTMO$_3$ is proved to significantly suppress the dissolution of Ni or Mn from the LNMO cathode material into the carbonate electrolyte, which is the leading degradation mechanism of high-voltage Co-free LIBs. Our atomistic explanations on material structure and property relation could also lead to the development of high-performance coating materials for other electrochemical systems.

## Methods
### Materials synthesis
*Pristine LNMO and BLNMO.* Pristine LNMO and BLNMO crystals were prepared by a polymer assisted sol-gel method[48]: 0.02 mol of nickel(II) acetate tetrahydrate, 0.06 mol of manganese(II) acetate tetrahydrate and 0.042 mol of lithium acetate dihydrate were dissolved in 150 mL of distilled water, followed by adding 0.12 mol of acrylic acid. The solution was stirred under 90 °C until dry gel was formed. Then

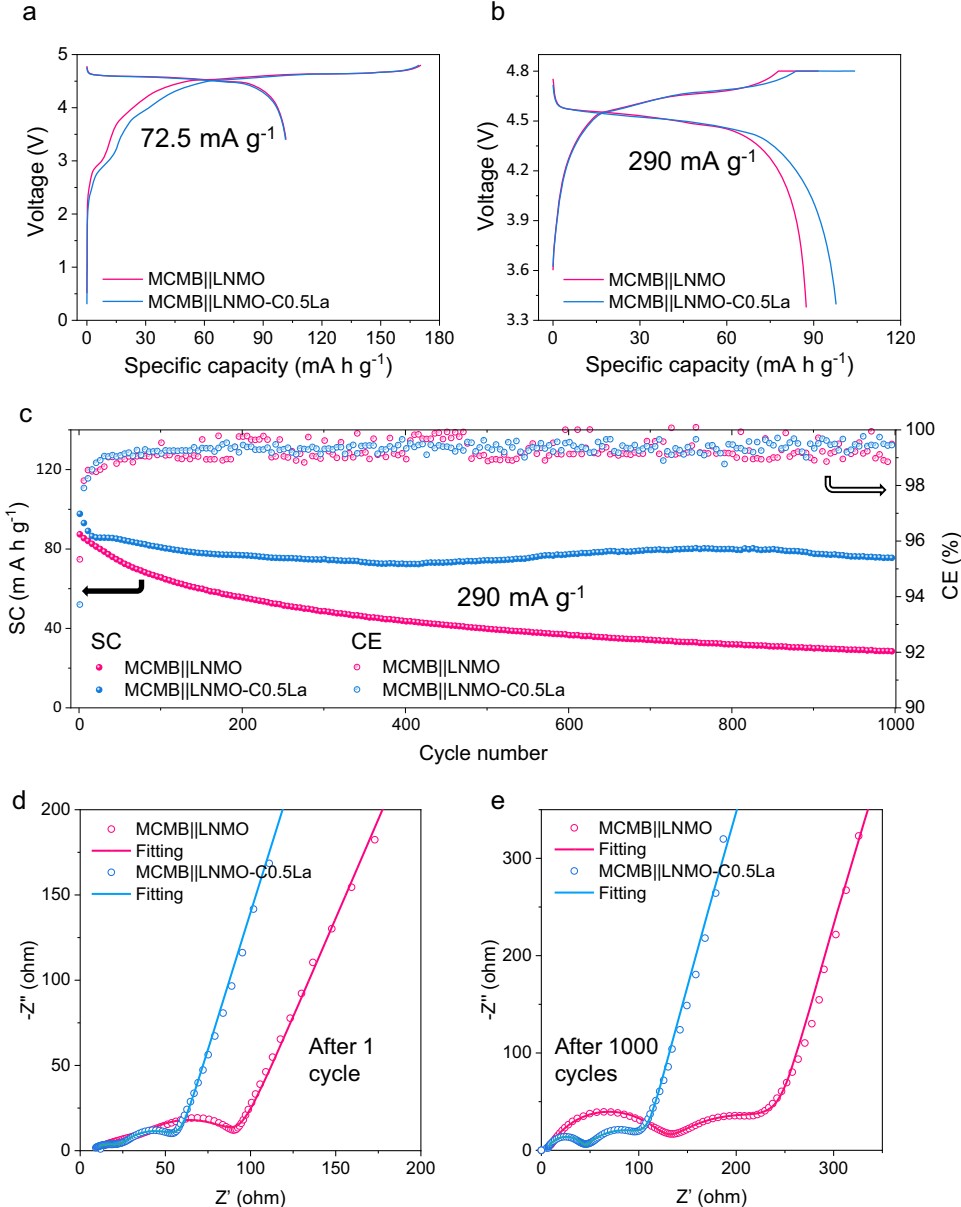

**Fig. 5 Electrochemical performance of MCMB∥LNMO and MCMB∥LNMO-C0.5La full cells at 26 °C. a** Initial charge/discharge profiles of the full cells at 72.5 mA g⁻¹. **b** The charge/discharge profiles of the full cells at 290 mA g⁻¹ (following constant current charge at 290 mA g⁻¹, an additional constant 4.8 V charge is applied until the current drops to 29 mA g⁻¹). **c** Long cycling performance of the full cells at a specific current of 290 mA g⁻¹ after initial five cycles at 72.5 mA g⁻¹. SC and CE represent specific capacity and Coulombic efficiency, respectively. **d** Nyquist plots of the full cells using LNMO and LNMO-C0.5La after an initial cycle at 72.5 mA g⁻¹. **e** Nyquist plots of the full cells using LNMO and LNMO-C0.5La after 1000 cycles at 290 mA g⁻¹. All the EIS spectra have been fitted using the equivalent circuit model shown in Supplementary Fig. 17, similar to literature[43]. The key fitting results and errors are listed in Supplementary Table 2.

the dry gel was heated at 400 °C for 3 h and grounded into fine powder using a mortar. Afterwards, the powder was calcinated at 750 °C for 8 h to get LNMO or 900 °C for 6 h to obtain BLNMO. All chemicals were purchased from Sigma-Aldrich unless otherwise stated.

*Doped LNMOs.* The preparation of doped LNMOs is similar to that of pristine LNMO, where certain amounts of lanthanum(III) nitrate hexahydrate, aluminum nitrate nonahydrate, gallium(III) nitrate hydrate, or yttrium(III) nitrate hexahydrate were introduced in the precursor solution to obtain pre-mixed precursor followed by being calcinated at 750 °C for 8 h.

*Coated LNMOs.* The coated LNMOs were prepared by a mixing and sintering process: 1 g of LNMO or BLNMO was dispersed in 10 mL of ethanol (99.8%) followed by the addition of required amount of nitrate salts, and the mixture was stirred under 40 °C until dry. Afterwards, the dry mixtures were re-sintered at 750 °C for 1 h to obtain the final products.

**Electrochemical characterization.** Charge/discharge tests were carried out using 2032 coin-cells. Each cell consisted of an LNMO based cathode and a Li counter electrode (99.9%, 0.5 mm thick, Guangdong Canrd Ltd) or a mesocarbon microbeads (MCMB, Guangdong Canrd Ltd) based anode separated by a polymer membrane (Celgard 2500, 25 μm thick, 55% porosity). To prepare the working electrode, active material, acetylene black and polyvinylidene fluoride (Solef PVDF 5130/1001, Solvay) were mixed with a weight ratio of 85:10:5, and N-methyl pyrrolidone (NMP) serving as the solvent to obtain a slurry. Then the slurry was coated on an aluminum foil (99.6%, 15 μm thick, Guangdong Canrd Ltd) for the cathode (coating thickness of 200 μm) or a copper foil (99.99%, 15 μm thick, Guangdong Canrd Ltd) for the anode (coating thickness of 90 μm). The electrodes were dried at 110 °C in vacuum for 12 h. After that, the electrodes were cut into round chips and compressed at a pressure of 10 MPa. The loading mass of the electrodes is 4 ± 0.5 mg cm⁻² for cathode and 1.8 ± 0.2 mg cm⁻² for anode. The N/P ratio for the coin-type full cell is around 1.11. 1 mol L⁻¹ solution of LiPF₆ dissolved in ethylene carbonate/diethyl carbonate (EC/DEC/DMC) (1/1/1, v/v/v)

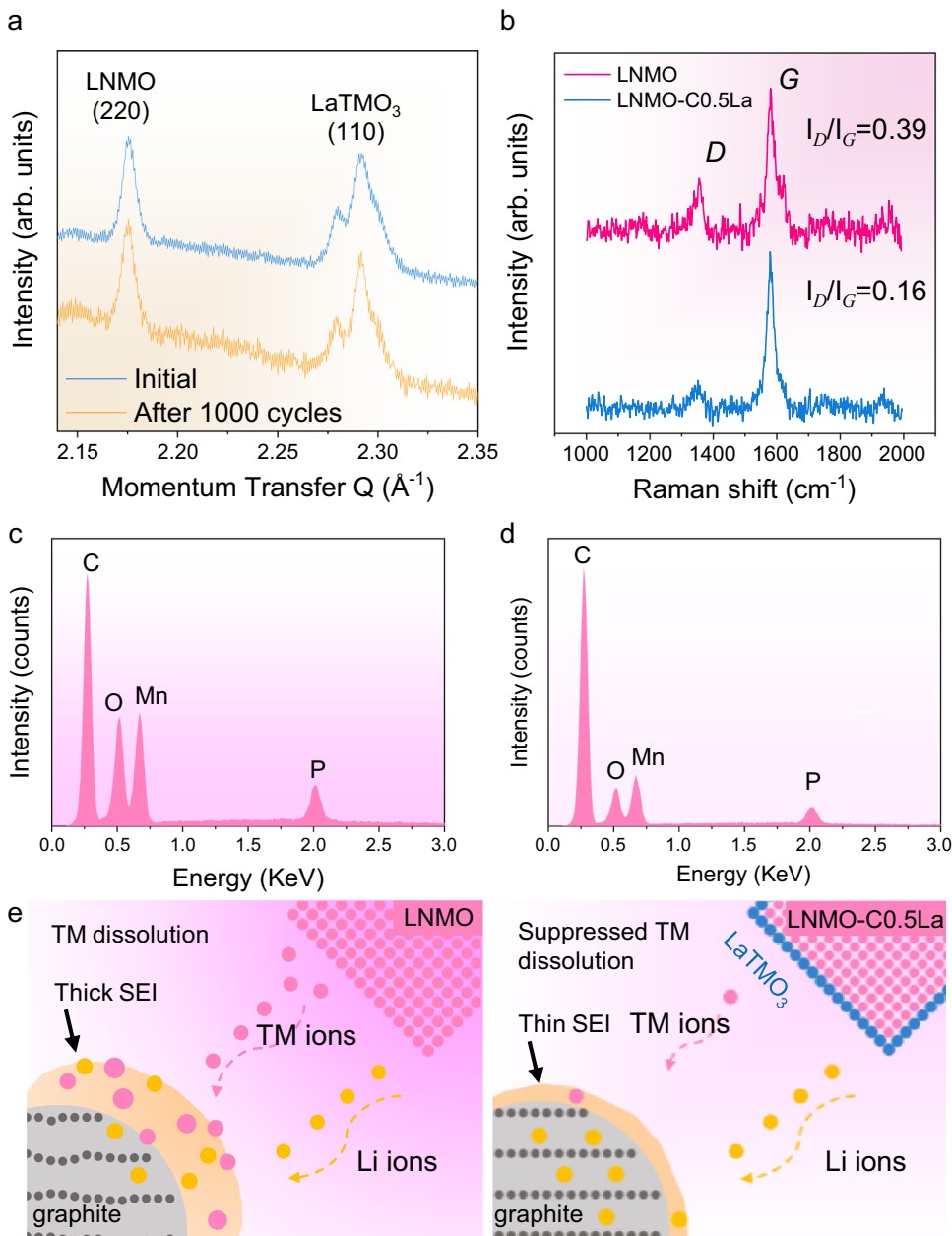

**Fig. 6 The role of LaTMO₃ surface protection on the durability of the 5 V-class LIBs. a** Ex situ post-mortem Synchrotron XRD measurement showing the LNMO (220) peak and the LaTMO₃ cubic (110) reflection for LNMO-C0.5La cathodes before and after cycling in the 3.4–4.8 V full cells. **b** Ex situ post-mortem Raman spectra of the graphite anodes harvested from cycled full cells using LNMO and LNMO-C0.5La cathodes. **c**, **d** Ex situ post-mortem EDS spectra of the cycled graphite anodes coupled with LNMO (**c**) and LNMO-C0.5La (**d**). **e** Schematic of the effect of the suppressed TM dissolution on the preservation of graphite anode. All the electrodes were collected from full cells at discharged state after 1000 cycles at 290 mA g⁻¹ under 26 °C.

with or without 0.2 mol L⁻¹ lithium difluoro(oxalato)borate (LiDFOB) was used as the electrolyte (60 μL of the electrolyte was used for each coin cell). The cells were assembled in an argon-filled glove box with oxygen and water levels below 0.5 ppm (MBRAUN). The charge–discharge tests were performed on a Neware battery testing system (CT-4008T) at 26 ± 2 °C in an air-conditioned room or at 45 ± 1 °C in a temperature chamber (WHL-25AB, Tianjing Taisite Instrument Co., Ltd). The potentiostatic EIS spectra were collected from an electrochemical workstation (Squidstat Plus, Admiral Instruments) in a frequency range from 1 MHz to 0.01 Hz (80 overall data points). The amplitude of the signal was 10 mV. The cells were at discharged state (0% SoC) and the OCV time applied before carrying out the EIS measurement was 1 h. The direct current bias was 0 V with respect to OCV.

**Materials characterization**. The crystalline phase of the samples was characterized by Synchrotron X-ray powder diffraction (λ = 0.07748 nm) radiation. The morphologies of the as-prepared samples were investigated by a field emission

scanning electron microscope (SEM, JSM-7100F, JEOL Ltd., Japan). The microstructure and local elemental distribution of the La-contained spinel were analysed by a cold field emission transmission electron microscope (TEM, Hitachi HF5000, Japan) fitted with spherical aberration corrector on the probe-forming lens systems. The XPS was carried out with monochromatic Al Kα excitation source. The calibration binding energy for C 1s was 284.8 eV. NEXAFS spectra were collected at the Soft X-ray beamline with TEY mode at the Australian Synchrotron. The deposition of Mn and Ni from cathode materials on Li electrodes was quantified by disassembling the half cells in the argon-filled glove box. The Li electrodes were exposed to air for 10 h and then dissolved in 50 mL HNO₃ solutions of 0.3 mol L⁻¹. The solutions were measured by Agilent 7700x inductively coupled plasma mass spectrometry (ICP-MS).

For the ex situ Raman measurements, the cells were dissembled in the argon-filled glove box. Subsequently, the graphite anodes were collected and washed with DMC. Then the anodes were transferred out and then dried under vacuum at 26 °C for 10 h. Finally, the electrodes were measured by a Raman microscope (Renishaw

inVia) with a laser wavelength of 514 nm. For the ex situ XRD measurement, the cathode powders were scratched from the electrodes in the glove box, filled in a sample holder and sealed with Kapton tapes. Thereafter, they were measured by Synchrotron X-ray powder diffraction. For the ex situ SEM and XPS measurements, the electrodes were loaded on the sample stages and sealed in sample tubes filled with Ar gas. When transferring the sample stages into the SEM and XPS instruments, the samples will be exposed to the air for a few seconds.

**Computational methods**. Spin-polarized DFT calculations were carried out by using the projector-augmented plane-wave (PAW) method as implemented in the Vienna Ab Initio Simulation Package (VASP), with planewave basis set and energy cutoff of 500 eV[49]. The exchange-correlation function was described by the Perdew–Burke–Ernzerhof generalized gradient approximation with the Hubbard $U$ corrections (PBE-GGA + $U$)[50], with $U–J$ value of 6.2 and 3.9 eV for Ni and Mn atoms, respectively[51]. For the Ni positions in LNMO, we considered Ni present on the surface of LNMO as well as in the interior of LNMO substrate (see Supplementary Fig. 1). The latter system is much lower in energy and thus would be more stable. The LNMO (111) substrate was modelled by a slab consisting of 10 atomic layers, with a vacuum space of 1.5 Å applied in the vertical direction. The Brillouin zone was sampled by $7 \times 7 \times 1$ uniform $k$-point mesh. With fixed cell parameters, the model structure was optimized for the ionic and electronic degrees using the convergence criteria of $10^{-4}$ eV for the electronic energy and $10^{-2}$ eV/Å for the forces on each atom, respectively. Grimme's semiempirical DFT-D3 scheme of dispersion correction was adopted to describe the van der Waals (vdW) interactions between the $LaNi_2O_3/LaMn_2O_3$ layer and the LNMO (111) surface[52]. The bottom 5 atomic layers of the substrate are frozen during the structure optimization to mimic a semi-infinite substrate.

**Reporting summary**. Further information on research design is available in the Nature Research Reporting Summary linked to this article.

## Data availability
The authors declare that all the relevant data within the paper and its Supplementary Information file or from the corresponding author upon reasonable request. The source data of Figs. 1d, 2a, d, 3a–d, 5a–e, 6a–d; Supplementary Figs. 1e, 2, 3, 8c, 10, 14, 15, 16, 18, 19, 20, 22, 23, 24,and 25 are provided as Source Data file. Source data are provided with this paper.

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

## Acknowledgements

The authors acknowledge the support from the Australian Government through the Australian Research Council's Laureate Fellowship (FL190100139) and Linkage Projects scheme (LP170100392), and from the Baosteel-Australia Joint Research and Development Centre (BA16011). This research was undertaken on the PD and SXR beamlines at the Australian Synchrotron, part of ANSTO. The authors acknowledge the facilities, and the scientific and technical assistance, of the Australian Microscopy & Microanalysis Research Facility at the Centre for Microscopy and Microanalysis, The University of Queensland and the computer resources provided by the NCI National Facility through the University of Wollongong Partner Share Scheme.

## Author contributions

X.Z., T.U.S., and L.W. conceived of the idea, designed the experiments, analyzed the data, and wrote the paper. X.Z., T.U.S., T.L., Y.H., N.C., F.H., K.O., B.C., Q.G., Z.C., and Y.D. carried out most of the characterizations and device optimizations. X.Y. and S.Z. performed the theoretical simulations. X.Z. and T.U.S. participated in the materials preparation, electrochemical characterization, and data analysis. T.U.S., T.L., and Q.G. carried out Synchrotron XRD characterizations and analyzed the results. B.C. carried out NEXAFS characterizations and analyzed the results. N.C., F.H., and K.O. did the HR-TEM characterizations of the materials. All authors commented on the manuscript.

## Competing interests

The authors declare no competing interests.
