## [Peer Review File · Nature Communications]

REVIEWER COMMENTS

Reviewer #1 (Remarks to the Author):

Comments:

In this work, an atomic wet layer growth approach of La(TM)O₃ (where TM \diamond Ni, Mn) monoatomic layer is utilized on LNMO cathode material in order to improve cycling stability of the LNMO cathode material. The authors report a Stranski–Krastanov growth of the monoatomic layer. Satisfactory electrochemical properties were demonstrated during electrochemical testing. The authors state that due to the atomic layer, there is limitation in transition metal dissolution during battery operation resulting in better cycle life (92% after 1000 cycles). Overall, the manuscript is well organized and can be accepted for publication after some revision.

1. It is necessary to provide a clear conclusion for this manuscript. Here, the authors have not mentioned in the summary whether the coated or doped had better performance and which concentration of La provided the best results. This must be added to the summary.
2. The authors state that 0.5% coating of La shows the best rate capability performance. Between the 4 different concentrations of La (0.3, 0.5, 0.7, and 1) what is the trade-off between the concentration and electrochemical performance? Is there any direct relation between these?

Reviewer #2 (Remarks to the Author):

A simple sol-gel method combined with the DFT calculations was applied to demonstrate the epitaxial layer of LaTMO₃ on the surface of LNMO cathode material. However, the experimental evidence is insufficient to prove such a thin layer is epitaxial growth instead of surface doping. Since the surface of LNMO usually does not have perfect crystal, so atomic defects or lattice deformation may appear on the surface. In this case, foreign atoms doping or substitution may be possible. Moreover, the proof of LaTMO₃ growth mode is also not sufficient.

So, the paper needs very significant improvement before acceptance for publication. My detailed comments are as follows:

1. It is suggested to supplement ABF-STEM images taken at the edge of LNMO-C0.5La and at the boundary between the spinel and the La-rich surface island.
2. It is suggested to measure the AFM images of LNMO in different states during the synthesis process in order to observe the surface roughness and distinguish the layer growth mode from the layer-island growth mode.
3. Whether LaNiO₃ or LaNi₂O₃ grown on the surface? At the beginning, LaNi₂O₃ was written, but the crystal structure given later is LaNiO₃, as shown in Fig. 1 G & J.
4. The peak shape of unprocessed LNMO in Figure 1F and S2 and S3 of the text is not consistent. For the same sample tested twice, the integrated peak shape should be the same. If it is the same sample, is there a problem with the integration or the sample?
5. It is suggested to supplement the quantitative results of EDS from surface layer to bulk phase of LNMO-C1.0La.
6. It is suggested that the impedance spectrum should be fitted.
7. In Fig 4 C&D, the Raman spectra of the graphite anode before cycle should be provided for comparison because the Raman spectra (insets) reveal increased ID/IG only, but not much change in FWHM after 1000 cycles. On the other hand, the electrochemical performance of LNMO and LNMO-C0.5La should be tested using the electrolyte without

additive such as LiDFOB. This is because LiDFOB is a forming layer additive, the good cycling life may be also benefiting from this additive.

8、 Why is the initial coulomb efficiency of the treated LNMO lower than that of the untreated LNMO during long cycling, as shown in Fig 3C and 3D. On the contrary, as shown in Fig. 5a, the coulomb efficiency of LNMO is lower than that of LNMO-C0.5La at 0.5C. How to explain this paradoxical results?

9、 It is suggested to add the cycle performance of untreated LNMO in Figure S13.

10、 What is the reason for the capacity increasing during cycling performance test for LNMO-C0.5La in Figure 3D and S15.

11、 The ionic radius of Y is smaller than La, so does Y dope into the lattice of LNMO in LNMO-C0.5Y? The rate performance of LNMO-C0.5Y seems to be better than that of LNMO-C0.5La. Why? Is the modification mechanism of LNMO-C0.5Y different from LNMO-C0.5La?

12、 Is there a problem with the caption of figure S5? Check it carefully.

13、 Some relevant literature should be consulted, such as
10.1021/acs.chemmater.6b00948, 10.1002/aenm.201701398,
10.1021/acs.chemmater.8b03764, 10.1007/s11426-020-9879-8,
10.1021/acsami.9b22358.

Reviewer #3 (Remarks to the Author):

The key result of the manuscript is the development of a simple mixing and calcination method to induce a self-limiting LaNiO₃ surface layer on LNMO, demonstrating it as a scalable method for improving capacity retention at high voltage operation of LNMO/graphite full cells. The results here show impressive capacity retention of 92% after 1000 cycles at relevant rates of 2C in full cells.

Transition metal migration does limit the successful commercialization of the high voltage spinels, such as LiMn_{0.5}Ni_{1.5}O₄. Various effects currently exist regarding the use of coatings, dopants, morphology and electrolyte additives to improve the capacity performance and retention. The main significance of this work is employing Stranski-Krastanov growth to obtain a self-limiting monoatomic thicknesses with a straighter forward mixing and calcination approach; as a result additional manufacturing steps are not required such as ALD coating etc...

The approach combines a DFT and XRD study is rigorous and identifies how the self-limited growth of the passive layer results in a monolayer thin metallic layer. This acts to improve the rate performance with limited impact on the underlying structure. They explore a range of growth and sintering conditions to optimize the cathode material. They employ STEM-EDX, NEXAFS and depth-profile XPS studies to confirm thin surface La³⁺ environments at the surface.

The resultant electrochemical studies do show for the full cells (where the impact of TM dissolution is most notable) improvements at relevant rates, including at elevated temperatures. Table S1 includes a wide literature review of similar studies for these systems. The authors provide post-cycling analysis of the separator and graphite anodes after cycling and confirm reduced Mn poisoning.

Overall this is a very thorough and promising study and is potentially suitable for the Nature Comm. audience. Before publication, the authors need to address the following:

1. Can the authors provide details of the full cells used in this study. I could only find

information on the 1/2 cells. This is important regarding the cell format (coin vs pouch; N:P) for evaluating in the field.

2. The addition of the LiDFOB additive needs to be clarified, since this additive has been shown to be successful in suppressing TM dissolution in NMC compounds eg. ACS Energy Mater 3 (2020) 695. The use of the additive makes it difficult to determine whether the passivation layer alone is critical. Currently, one could argue that the LiDOB is significantly contributing to suppressing the TM dissolution.

3. For the 1/2 cells, the authors could provide chemical analysis of the Li counter electrodes to see if the same TM dissolution occurs in that case - especially the difference in Mn and Ni ratios.

4. For the full cells the specific capacity shows large variations with cycling, can the authors explain this? This would not be suitable for end use applications and limits the impact of this study. The effects are very pronounced at 45C. The increase in capacity over time suggests a combination of effects are occurring e.g. side reactions.

5. Is there any reason why the Mn dissolution is suppressed but the Ni dissolution is not impacted as much.

Reviewer #1 (Remarks to the Author):

Comments:

In this work, an atomic wet layer growth approach of La(TM)O₃ (where TM \diamond Ni, Mn) monoatomic layer is utilized on LNMO cathode material in order to improve cycling stability of the LNMO cathode material. The authors report a Stranski–Krastanov growth of the monoatomic layer. Satisfactory electrochemical properties were demonstrated during electrochemical testing. The authors state that due to the atomic layer, there is limitation in transition metal dissolution during battery operation resulting in better cycle life (92% after 1000 cycles). Overall, the manuscript is well organized and can be accepted for publication after some revision.

1. It is necessary to provide a clear conclusion for this manuscript. Here, the authors have not mentioned in the summary whether the coated or doped had better performance and which concentration of La provided the best results. This must be added to the summary.

Response: We appreciate the recognition and valuable suggestions from the reviewer.

Introducing La by the coating way leads to better electrochemical performance compared to mixing La by the doping method. At the origin of the phenomenon of our coating procedure is the insolubility of La in LNMO. Hence for the coated material, the essential fraction of La can be found in the surface region after annealing as shown in the NEXAFS data presented in the article. The insolubility also representing an important diffusion barrier leads to a lower crystal quality for the doped sample due to the presence of big La ions during the crystallization of LMNO. The same properties (insolubility and poor bulk diffusion of La in LNMO) combined with epitaxial compatibility favor in the coating procedure a homogeneous surface coverage with

La on the previously formed high quality crystals. Among the different concentrations of La coating, our experiments revealed that LNMO-C0.5La with 0.5at% La coating shows the best performance, and more importantly the best cycling stability in the full batteries, because of the good coverage of LaTMO₃ wetting layer. Less amount of La coating (LNMO-C0.3La) leads to insufficient coverage of the LNMO materials, while excess amount of La (LNMO-C0.7La) tends to form thicker islands, hindering the charge transfer. Following the suggestion from the reviewer, we have added a summary discussing our key findings in the summary part of the revised manuscript, as listed below;

“The immiscibility of La in LNMO, energetical advantage of La-O terminations, together with the lattice mismatch led to the Stranski–Krastanov growth of LaTMO₃ wetting layer when La is introduced by the coating approach. In consistence with the growth mode, quantitative X-ray methods confirm that 0.5 at% La is required for the appropriate surface coverage of LaTMO₃ wetting layer on the LNMO octahedra. This explains the improved electrochemical performance of the LNMO-C0.5La compared with pristine LNMO, La doped LNMO (LNMO-D0.5La), and LNMOs coated by insufficient (LNMO-C0.3La) or excessive LaTMO₃ (LNMO-C0.7La).”

2. The authors state that 0.5% coating of La shows the best rate capability performance. Between the 4 different concentrations of La (0.3, 0.5, 0.7, and 1) what is the trade-off between the concentration and electrochemical performance? Is there any direct relation between these?

Response: We appreciate this excellent comment. The LNMOs intrinsically deliver excellent capability in half-cells attributing to the 3-dimentional spinel framework which offers fast ion

diffusion and experiences a minimum of structural fatigue over cycling. The main purpose of the ultrathin LaTMO₃ coating is to alleviate the capacity drop of LNMO in full battery by suppressing the TM dissolution. Quantitative X-ray methods led us to conclude 0.5at% La is required for a monolayer thickness of LaTMO₃ coating. Due to the monolayer thickness and electron conducting nature of our LaTMO₃ coating, LNMO-C0.5La with 0.5at% La coating shows uncompromised rate performance. But when excess amount of La is introduced, the additional appearance of LaTMO₃ islands has no further beneficial contribution to the protection of the LNMO but presents a resistance to the charge transfer properties and also raises the material costs. We understand the misunderstanding from the reviewer, which may also confuse readers as there are too many control samples. As both 0.7at% and 1at% are at excess conditions, for simplification, we have redone the electrochemical measurements at strictly controlled electrode processing and removed the comparison for 1at% La. In response, we have clarified this in the revised manuscript:

“All the LNMOs show excellent rate capability (Fig. 3B) and long cycling stability in half cells with LiDFOB electrolyte additive (Fig. 3C). The excellent half-cell performance is attributed to the 3-dimensional spinel framework which offers fast ion diffusion and experiences a minimum of structural fatigue over cycling (36-39). As predicted, the ultrathin LaTMO₃ coating does not compromise the rate performance of LNMO due to the monolayer thickness and electronic conducting nature. However, excessive amount of La (LNMO-C0.7La) slightly affects the rate performance (Supplementary Figure 15), which may be due to the thickening of relaxed LaTMO₃ islands that hinder the charge transfer.”

Reviewer #2 (Remarks to the Author):

A simple sol-gel method combined with the DFT calculations was applied to demonstrate the epitaxial layer of LaTMO₃ on the surface of LNMO cathode material. However, the experimental evidence is insufficient to prove such a thin layer is epitaxial growth instead of surface doping. Since the surface of LNMO usually does not have perfect crystal, so atomic defects or lattice deformation may appear on the surface. In this case, foreign atoms doping or substitution may be possible. Moreover, the proof of LaTMO₃ growth mode is also not sufficient.

So, the paper needs very significant improvement before acceptance for publication. My detailed comments are as follows:

1、 It is suggested to supplement ABF-STEM images taken at the edge of LNMO-C0.5La and at the boundary between the spinel and the La-rich surface island.

Response: We appreciate the valuable comments given by the reviewer and have conducted additional ABF-STEM characterization at the edge of LNMO-C0.5La and at the boundary between the spinel and the La-rich surface islands as shown in Supplementary Figure 12. The discontinued outer layer and the connection between bulk and surface phases are consistent with the HAADF-STEM observations.

Meanwhile, it should be noted that in the case of thick particles (over 200 nm), STEM imaging is less capable to characterize the monolayer. Instead, the islanded secondary phase and its epitaxial interface can be directly imaged by STEM, which evidence the growth mode. To this end, we quantified the amount of material contributing to the scattering process using

quantitative X-ray methods. Independently of the detailed surface sensitive spectroscopic methods and electron microscopies this led us to conclude on its monolayer thickness (Fig. 2 A, B in particular). These X-ray methods are well established serving as a basic tool in many disciplines, and we cited accordingly. For clarity, we have further explained in more detail in the supplementary materials our approach to derive these quantities from the relative intensities of the diffraction peaks (Eqs. 1 to 5 in the Materials and Methods part). More details in a general sense of such derivation can also be found in some textbooks (e.g., B. *Cullity Elements of X-ray Diffraction, Addison-Wesley Publishing Company, Reading, Massachusetts, 1956. p.388-391*). As one of the corresponding authors, Dr. Tobias U. Schüllli from the European Synchrotron, has a strong background in epitaxial growth and X-ray analysis, we have applied these analysis methods to understand the new structures in this work, revealing the monolayer nature of the La coating in the LMNO cathode materials.

In response, we have added related results and discussion in the revised manuscript:

“The annular bright-field scanning transmission electron microscopy (ABF-STEM) images taken at the edge of LNMO-C0.5La and at the boundary between the spinel and the La-rich surface islands are shown in Supplementary Figure 12. In combination with the XRD analysis, the crystal growth mode of wetting and islanding can be further verified.”

Supplementary Figure 12. ABF-STEM images taken at the edge of LNMO-C0.5La and at the boundary between the spinel and the La-rich surface island for BLNMO-C1.0La.

2、 It is suggested to measure the AFM images of LNMO in different states during the synthesis process in order to observe the surface roughness and distinguish the layer growth mode from the layer-island growth mode.

Response: We appreciate the comment and have applied the AFM to observe the surface roughness and distinguish the layer growth mode from the layer-island growth mode. However, this is very challenging as the substrate is not a flat two-dimensional film. The nano/microcrystals are polyhedral and the surfaces of this polyhedral are not ideally flat. As shown in Figure R1, the layer growth mode samples (LNMO-C0.5La) cannot be distinguished from the pristine samples with AFM methods due to the nature of the coating that is not interfering with the topography of the already present facets. The values varied greatly from facet to facet and particles to particles, which cannot give reliable comparison for different samples. From our experiments, quantitative X-ray methods are more suited to understand the growth process.

Figure R1. AFM images of LNMO (A) and LNMO-C0.5La (B) particles.

3、 Whether LaNiO_3 or LaNi_2O_3 grown on the surface? At the beginning, LaNi_2O_3 was written, but the crystal structure given later is LaNiO_3 , as shown in Supplementary Figure 1 G & J.

Response: We are sorry that we haven't provided very clear explanation about the new La-based phases on the surface. Our XRD experiments and other characterizations revealed that there is an optimal La-coating to form the desirable one atomic monolayer of the growing LaNiO_3 (or more generally LaTMO_3) structure.

With the epitaxial surface presence of La, pseudomorphically strained to the lattice parameter of LNMO a precise crystallographic determination of the impact of this structure on the underlying

crystal is not possible from powders, although it is known from surface diffraction that such influences are common in most materials. We have thus used DFT in order to simulate how this surface structure “connects” to the underlying lattice. The surface unit cell used for this treatment reaches into the first transition metal layer and is hence referred to as LaNi_2O_3 . Extending it further would be possible, implying also a renaming as a function of the changing stoichiometry, with La becoming less and less prominent in the sum formula, the more one extends into the bulk.

Accordingly, we added a description in the revised manuscript:

“Simulation optimization reveals that such atomic layer consisting of La: Ni: O in a ratio of 1: 2: 3, where the surface unit cell reaches into the first TM layer and is hence referred to as LaNi_2O_3 , indicating a high degree of integration between surface structure and the underlying lattice.”

4、 The peak shape of unprocessed LNMO in Figure 1F and S2 and S3 of the text is not consistent. For the same sample tested twice, the integrated peak shape should be the same. If it is the same sample, is there a problem with the integration or the sample?

Response: We are sorry that there might be some misunderstanding from the reviewer. The presented datasets are identical for the pristine LNMO. The size of the zoomed regions is different however, as in Figure 1F it focuses on the apparition of the perovskite LaTMO_3 extra phase and span over the angular range from the LNMO (220) peak up to the perovskite LaTMO_3 main peak in order to see subtle difference also in the structure of this extra phase. The X-axis is here indicated to reach from 2.15 to 2.33 \AA^{-1} .

The Supplementary Figures 2 and 3 show the impact of different elements used in a doping or coating procedure. The zoomed region represents here the low-angle region “typical” for metal oxide Bragg peaks in order to give better visibility to eventual extra phases. The X-axis is here as well indicated to range from 1.5 to 3 Å⁻¹, a range about 10 times as high as in Fig 1F and including some very intense peaks from the LNMO.

We have further clarified the figure caption description by adding the Supplementary Figure legends 2 and 3, and by adding frames to Figs. 1E and 1H to indicate clearly the zoomed region magnified in Figs. 1F and 1I.

5、 It is suggested to supplement the quantitative results of EDS from surface layer to bulk phase of LNMO-C1.0La.

Response: The quantitative EDS analyses have been conducted and the results of from surface layer to bulk phase of LNMO-C1.0La have been provided in Supplementary Table 2.

Correspondingly, we have added a description in the revised manuscript:

“The compositional difference between the island and the bulk has been further verified by the EDS quantitative analysis (Supplementary Figure 10).”

	Ni K at%	Mn K at%	La L at%	O K at%	C K at%
Point 1 (island)	5.03	18.58	5.70	33.56	37.13
Point 2 (bulk)	5.08	17.45	0	50.63	26.84
Overall	4.82	19.68	0.17	57.38	17.95

Supplementary Figure 10. EDS quantitative analysis of BLNMO-C1.0La from the surface island to the bulk phase.

6、 It is suggested that the impedance spectrum should be fitted.

Response: The impedance spectra have been remeasured on a Squidstat Plus electrochemical workstation with strictly controlled conditions and the data has been fitted accordingly. A related explanation has been added in the revised manuscript “According to the fitting results (Supplementary Figure 20), the surface film resistance (R_{sf}) and charge transfer resistance (R_{ct}) of the standard LNMO/MCMB cell is 1.4 and 1.7 time higher than those of LNMO-C0.5La/MCMB cell after first cycles. Remarkably, the differences are enlarged to 3.1 and 2.0 times after 1000 cycles.”

Supplementary Figure 20. EIS results of the full batteries. (A) Equivalent circuit model for EIS data fitting (57). (B) Nyquist plots and fittings of the full cells using LNMO and LNMO-C0.5La after 1 cycle. (C) Key results obtained from fitting.

7、 In Fig 4 C&D, the Raman spectra of the graphite anode before cycle should be provided for comparison because the Raman spectra (insets) reveal increased ID/IG only, but not much change in FWHM after 1000 cycles. On the other hand, the electrochemical performance of LNMO and LNMO-C0.5La should be tested using the electrolyte without additive such as LiDFOB. This is because LiDFOB is a forming layer additive, the good cycling life may be also benefiting from this additive.

Response: Thanks for the valuable suggestion. LiDFOB is found as a necessary component in the test of high-voltage LNMO cathode materials. It is important to state that we compare here differently prepared LNMO cathode materials under identical conditions using LiDFOB as one component that does not differ between the different samples.

Nevertheless, as shown in Supplementary Figure 14, we have tested the LNMOs without the addition of LiDFOB. In fact, the advantage of LNMO-C0.5La is much more evident in the standard electrolyte without LiDFOB. The capacity drop of LNMO is significant after 200 cycles while LNMO-C0.5La remains stable before 500 cycles. After 1000 cycles, the capacity retention ratios of LNMO and LNMO-C0.5La are 36.8% and 51.9%, respectively. Notably, the capacity of the electrodes can be recovered by disassembling the cells and reassembling the electrodes into new cells with the LiDFOB-contained electrolyte, indicating the capacity drop is due to the decomposition of the standard electrolyte.

Supplementary Figure 14. Cycling performance of LNMO and LNMO-C0.5La in half cells at 725 mA g⁻¹. LNMO-C0.5La shows improved cycling stability in the standard electrolyte without

LiDFOB. The capacity drop of LNMO is significant after 200 cycles while LNMO-C0.5La remains stable before 500 cycles. After 1000 cycles, the capacity retention ratios of LNMO and LNMO-C0.5La are 36.8% and 51.9%, respectively. Notably, the capacity of the electrodes can be recovered by disassembling the cells and reassembling the electrodes into new cells with the LiDFOB-contained electrolyte, indicating the capacity drop is due to the decomposition of the standard electrolyte.

Correspondingly, we have further clarified this in the revised manuscript:

“To verify the effectiveness of the surface passivation, the electrochemical performance of the untreated and treated LNMOs has been carefully compared under the same experimental conditions including the use of LiDFOB as an electrolyte additive (35), which is found necessary for the extended cycling of LNMOs (Supplementary Figure 14).”

8、 Why is the initial coulomb efficiency of the treated LNMO lower than that of the untreated LNMO during long cycling, as shown in Fig 3C and 3D. On the contrary, as shown in Supplementary Figure 5a, the coulomb efficiency of LNMO is lower than that of LNMO-C0.5La at 0.5 C. How to explain this paradoxical results?

Response: We are sorry that we have not clarified the initial coulomb efficiency in the manuscript. Initially, Fig 3C starts from 70th cycle (after the rate performance test) and the Supplementary Figure 5a shows a typical charge-discharge curve (the fifth cycle). Accordingly, we have redone the electrochemical measurements. The initial charge-discharge curves for the half-cells have been provided in Fig. 3a and supplementary Figure 15a. The initial coulombic efficiencies are 77.8%, 79.2% and 80.4% for LNMO, LNMO-D0.5La, and LNMO-C0.5La in the

half cells. The initial charge-discharge curves for the full batteries have been provided in Supplementary Figure 16a and Supplementary Figure 17a. The initial coulombic efficiencies for the full batteries are around 60%. It should be note that in the case of coin cells, the parasitic reactions, which determine the initial coulombic efficiency, can be varied especially the full batteries, since for the limited area of electrode pieces, the evenness of the electrodes and the distribution of electrode components (active materials, carbon black, and binder) cannot ideally be the same.

In response to the reviewer's comment, we have added the initial charge-discharge curves in Fig. 3a, Supplementary Figure 15a, Supplementary Figure 16a and Supplementary Figure 17a.

Related discussions have been added in the revised manuscript:

“The initial coulombic efficiencies are 77.8%, 77.6% and 80.4% for LNMO, LNMO-D0.5La, and LNMO-C0.5La, respectively, indicating the LaTMO₃ coating can reduce the parasitic reactions between electrodes and electrolyte.”

“The charge/discharge curves of the full cells are illustrated in Supplementary Figure 16, which give initial Coulombic efficiency of around 60%.”

9、 It is suggested to add the cycle performance of untreated LNMO in Figure S13.

Response: The cycle performance of untreated LNMO has been added in Figure S13 for comparison, which is Supplementary Figure 17 in the revised version.

10、 What is the reason for the capacity increasing during cycling performance test for LNMO-C0.5La in Figure 3D and S15.

Response: The phenomena of capacity increasing can be explained by the lack of formation process for laboratory coin-cell studies. The formation process is one of the most important steps in the manufacturing process of lithium-ion batteries, during which electrolyte is added to the cell and then diffuses and completely wets the pores of the electrodes. Normally the process of liquid electrolyte diffusion into the porous electrode films takes several days or weeks at elevated temperatures (*Journal of Energy Storage* 2019, 26, 101034; *Journal of Power Sources* 2017, 342, 846-852). In laboratory coin-cell studies, there is no particular formation process before charge-discharge tests, the increment of capacity is believed to be related with the wetting and activation of electrodes during tests (Figure R2). Moreover, since our long cycling tests have been run at a relatively high current density (more cycles at a fixed timeframe) and the addition of 0.2 M LiDFOB further increases the concentration of the electrolyte, the capacity increasing can be evident if the TM dissolution driven battery failure is suppressed.

Accordingly, we clarified this in the revised manuscript:

“Since the long cycling tests have been measured at a relatively high current density and the addition of 0.2 M LiDFOB increases the concentration of the electrolyte, it could take more cycles for the complete electrolyte wetting. This explains the slight capacity increment of LNMO-C0.5La over cycling, where the TM dissolution driven battery failure is suppressed.”

Fig. R2 Typical cycling behavior of MCMB anode in a half cell.

11、 The ionic radius of Y is smaller than La, so does Y dope into the lattice of LNMO in

LNMO-C0.5Y? The rate performance of LNMO-C0.5Y seems to be better than that of LNMO-C0.5La. Why? Is the modification mechanism of LNMO-C0.5Y different from LNMO-C0.5La?

Response: According to our experiments, immiscibility has not been found for the alternative Al, Ga, and Y. In other words, Y is doped into LNMO lattice along sintering, other than forming a Y-coating on the LNMO surface. The rate performance of LNMO-C0.5Y seems to be better than that of LNMO-C0.5La in the half-cell testing, which may be explained by the doping effect of Y, like other Y-doped cathode materials (J. Electrochem. Soc. 2016, 163 A766, Electrochimica Acta 2010, 55, 28, 8815-8820). Although Al, Ga, and Y have been introduced in the same way as La, their miscibility behavior result in different modification mechanisms. Al, Ga, and Y tend

to dope into the LNMO lattice, while the immiscible La interact with the surface lattice of LNMO to form surface specific protective phase.

Correspondingly, we further explained this in the revised manuscript, as highlighted in red:

“Although Al, Ga, and Y have been introduced in the same way with La, these elements tend to dope into the LNMO lattice because of their miscibility with LNMO. As a result, LNMOs treated with these alternative ions show much less improvement of cycling stability in full batteries (Supplementary Figure 17).”

12、 Is there a problem with the caption of figure S5? Check it carefully.

Response: We have corrected this typo in the revised version.

13、 Some relevant literature should be consulted, such as

10.1021/acs.chemmater.6b00948,10.1002/aenm.201701398, 10.1021/acs.chemmater.8b03764,
10.1007/s11426-020-9879-8, 10.1021/acsami.9b22358.

The first of these papers looks indeed excellently matching

Response: Thanks for the suggestion. These papers are really relevant to our research which have been cited in the revised manuscript, as highlighted in red.

Reviewer #3 (Remarks to the Author):

The key result of the manuscript is the development of a simple mixing and calcination method to induce a self-limiting LaNiO_3 surface layer on LNMO, demonstrating it as a scalable method for improving capacity retention at high voltage operation of LNMO//graphite full cells. The results here show impressive capacity retention of 92% after 1000 cycles at relevant rates of 2C in full cells.

Transition metal migration does limit the successful commercialization of the high voltage spinels, such as $\text{LiMn}_{0.5}\text{Ni}_{1.5}\text{O}_4$. Various effects currently exist regarding the use of coatings, dopants, morphology and electrolyte additives to improve the capacity performance and retention. The main significance of this work is employing Stranski-Krastanov growth to obtain a self-limiting monoatomic thicknesses with a straighter forward mixing and calcination approach; as a result additional manufacturing steps are not required such as ALD coating etc...

The approach combines a DFT and XRD study is rigorous and identifies how the self-limited growth of the passive layer results in a monolayer thin metallic layer. This acts to improve the rate performance with limited impact on the underlying structure. They explore a range of growth and sintering conditions to optimize the cathode material. They employ STEM-EDX, NEXAFS and depth-profile XPS studies to confirm thin surface La^{3+} environments at the surface.

The resultant electrochemical studies do show for the full cells (where the impact of TM

dissolution is most notable) improvements at relevant rates, including at elevated temperatures. Table S1 includes a wide literature review of similar studies for these systems. The authors provide post-cycling analysis of the separator and graphite anodes after cycling and confirm reduced Mn poisoning.

Overall this is a very thorough and promising study and is potentially suitable for the Nature Comm. audience. Before publication, the authors need to address the following:

1. Can the authors provide details of the full cells used in this study. I could only find information on the 1/2 cells. This is important regarding the cell format (coin vs pouch; N:P) for evaluating in the field.

Response: We appreciate the recognition and valuable comments given by the reviewer.

Accordingly, we have redone the electrochemical measurements at strictly controlled conditions including electrode loading and compression pressure. The N/P ratio is roughly calculated as $\frac{325 \text{ mAh g}^{-1} \times 0.45}{130 \text{ mAh g}^{-1}} = 1.11$ based on an anode capacity of 325 mAh g⁻¹ (Fig. R2) and a cathode capacity of 130 mAh g⁻¹. Correspondingly, we have provided the details of the full cells in the revision:

“Then the slurry was coated on an aluminum foil (99.6%, 15 μm thick, Guangdong Canrd Ltd) for the cathode (coating thickness of 200 μm) or a cooper foil (99.99%, 15 μm thick, Guangdong Canrd Ltd) for the anode (coating thickness of 90 μm).”

“The loading mass of the electrodes is $4 \pm 0.5 \text{ mg cm}^{-2}$ for cathode and $1.8 \pm 0.2 \text{ mg cm}^{-2}$ for anode to give a N/P ratio of *ca.* 1.11”.

2. The addition of the LiDFOB additive needs to be clarified, since this additive has been shown to be successful in suppressing TM dissolution in NMC compounds eg. ACS Energy Mater 3 (2020) 695. The use of the additive makes it difficult to determine whether the passivation layer alone is critical. Currently, one could argue that the LiDFOB is significantly contributing to suppressing the TM dissolution.

Response: We appreciate this valuable comment which is relevant to the Comment 7 of the Reviewer 2. As mentioned, we would like to underline that we compared our differently prepared and coated LNMO cathode materials using LiDFOB as additive in an identical manner. The significant differences in cycling stability and TM dissolution are observed between samples that all contain the LiDFOB additive. In fact, as shown in Supplementary Figure 14, the advantage of LNMO-C0.5La is more evident in the standard electrolyte without LiDFOB.

3. For the 1/2 cells, the authors could provide chemical analysis of the Li counter electrodes to see if the same TM dissolution occurs in that case - especially the difference in Mn and Ni ratios.

Response: Thanks for this helpful suggestion. We have conducted additional ICP measurements for the Li counter electrodes and used the data to replace the previous ICP results collected from soaked graphite anodes for better accuracy. First, ICP-MS has been employed to analysis the solutions, which is more suitable than ICP-OES for the measurement of low-concentration solution (around or below 1 ppm). Second, Li counter electrodes can be completely dissolved in acid without the concern of losing the chemical species, while the graphite electrodes cannot be

dissolved. In addition, the half cells have been disassembled at charged state to make sure that the TMs were deposited on the Li foils. As shown in supplementary Figure 25, the ratios of deposited Ni are lower than Mn for all the conditions. The suppression of TM dissolution is also significant by comparing LNMO and LNMO-C0.5La under all test conditions (with or without LiDFOB).

Supplementary Figure 25. Suppressed TM dissolution enabled by cathode surface passivation. Calculated ratios of deposited Mn (A) and Ni (B) with respect to the Mn and Ni in the cathode materials by harvesting the Li metal anodes at charged state. Corresponding digital photographs of the Li electrodes. At charge state, the dissolved Mn and Ni are expected to be deposited on the Li foil:

Cathode

Anode

4. For the full cells the specific capacity shows large variations with cycling, can the authors explain this? This would not be suitable for end use applications and limits the impact of this study. The effects are very pronounced at 45C. The increase in capacity over time suggests a combination of effects are occurring e.g. side reactions.

Response: Thanks for this valuable comment which is relevant to the Comment 10 of the Reviewer 2.

The phenomena of capacity increasing are related with the electrolyte wetting, which usually takes several days or weeks at elevated temperatures (Journal of Energy Storage 2019, 26, 101034; Journal of Power Sources 2017, 342, 846-852). In laboratory coin-cell studies, there is no particular formation process before charge-discharge tests, the increment of capacity can be related with the wetting of electrode during tests rather than the side reactions, as the side reactions are more likely to reduce the capacity. Moreover, since our long cycling tests have been run at a relatively high current density (more cycles at a fixed timeframe) and the addition of 0.2 M LiDFOB further increases the concentration of the electrolyte, the capacity increasing can be evident if the TM dissolution driven battery failure is suppressed. For end-user applications, the particular formation process could be a topic for industrial R&D, which is out

of the scope of this study. In fact, we are working with industrial partners on demonstrating the technology.

The capacity increase of BLNMO-C0.1La over time at 45 °C could be related with the improved kinetics from several aspects. First, the elevated temperature reduces the viscosity of the electrolyte, improving the ionic mobility. Second, the elevated temperature also benefits the solid diffusion of Li^+ in LNMO and MCMB, especially for the BLNMO-C0.1La with larger particle size. The increased reaction kinetics help the electrodes to quickly approach their theoretical capacities. However, higher temperature also stimulates the interfacial reactions and TM dissolution. For the unprotected BLNMO, the TM deposits on the anode part are expected to rapidly consume the active Li, eliminating the possibility of on-going capacity activation. We have redone the electrochemical measurements. The faster capacity increase suggests that the electrolyte wetting is related with the temperature.

In response to this comment, two sentences have been added in the revision to further explain the phenomena:

“Since the long cycling tests have been measured at a relatively high current density and the addition of 0.2 M LiDFOB increases the concentration of the electrolyte, it could take more cycles for the complete electrolyte wetting. This explains the slight capacity increment of LNMO-C0.5La over cycling, where the TM dissolution driven battery failure is suppressed”.

“The extended capacity increase could be related to the bigger crystallite size as well as the effective surface protection.”

5. Is there any reason why the Mn dissolution is suppressed but the Ni dissolution is not impacted as much.

Response: Following your suggestion, we have provided chemical analysis of the Li counter electrodes to better understand the TM dissolution behavior. The ICP results suggest that the dissolution of both Mn and Ni is significantly suppressed. The ratio of deposited Ni is in fact slightly lower than that of Mn. The relevant details have been provided in the revision,

“The amount of TM dissolution has been further quantified by inductively coupled plasma mass spectrometry (ICP-MS) analysis (Supplementary Figure 25), which confirms that the ultrathin LaTMO₃ wetting layer pronouncedly reduces the TM deposition on Li electrodes at different conditions”.

REVIEWERS' COMMENTS

Reviewer #2 (Remarks to the Author):

A simple sol-gel method combined with the DFT calculations was applied to demonstrate the epitaxial layer of LaTMO₃ on the surface of LNMO cathode material. Satisfactory electrochemical properties were demonstrated during electrochemical testing.

The authors have made sufficient modifications according to the modification comments, and I suggest that this paper be accepted without further modification.

Reviewer #3 (Remarks to the Author):

The authors have adequately addressed my concerns. The addition of the ICP-MS studies and the role of the LiDFOB additive have strengthened their conclusions. Meanwhile, the manuscript has overall been improved by clarifying the authors concerns and by better summarizing their key findings regarding the LaTMO₃ wetting layers impact on the electrochemical performance. Overall I believe it is suitable for publication in its current form.